# Exploring Adhesive Performance in Horseshoe Bonding Through Advanced Mechanical and Numerical Analysis

**DOI:** 10.3390/biomimetics10010002

**Published:** 2024-12-24

**Authors:** C. M. C. Ferreira, B. D. Simões, E. A. S. Marques, R. J. C. Carbas, L. F. M. da Silva

**Affiliations:** 1Institute of Science and Innovation in Mechanical and Industrial Engineering (INEGI), Rua Dr. Roberto Frias, 4200-465 Porto, Portugal; bsimoes@inegi.up.pt (B.D.S.); rcarbas@inegi.up.pt (R.J.C.C.); 2Department of Mechanical Engineering, Faculty of Engineering, University of Porto, Rua Dr. Roberto Frias, 4200-465 Porto, Portugal; emarques@fe.up.pt (E.A.S.M.); lucas@fe.up.pt (L.F.M.d.S.)

**Keywords:** equine industry, horseshoeing, acrylic adhesives, numerical modelling

## Abstract

Despite technological advancements in various industries, the equine sector still relies on old methods like horseshoeing. Although traditional, the industry is dynamic and well-funded. Therefore, there is a need to modernize these methods with more reliable and less invasive solutions for attaching horseshoes to horse hooves. There are currently several commercial adhesive solutions in the market specifically tailored to this application. In this work, the mechanical properties of two acrylic adhesives were characterized under quasi-static conditions. In the characterization process, tensile, shear, and fracture properties were determined. Subsequently, in-joint behavior was assessed using single-lap joints (SLJ) for both similar and dissimilar adherends. The adherends’ materials included steel (St), aluminum (Al), and horse hoof wall (HW), and the following adherend combinations were tested: St–St, Al–Al, and St–HW. A numerical model of similar joints was developed and validated based on experimental results. After its validation, the next steps are the modelling of the real joint and its simulation by considering realistic loading conditions in order to determine the weakest points of the joint. This exploratory study seeks to establish a foundation for investigating alternative adhesive solutions that could address the limitations identified in the solutions studied in this paper.

## 1. Introduction

Over the past few decades, most industrial sectors have experienced exponential technological advancement, largely driven by the development of new materials and methods. Nevertheless, certain traditional sectors continue to employ longstanding practices. The equestrian industry is one of those sectors, where horseshoe application has remained consistent with methods dating back to medieval times [1].

Despite its traditional methods, the equestrian sector attracts substantial investment and is a dynamic, challenging, and competitive industry. In Europe, it represents a market valued at approximately 100 billion euros a year [2], which, proportionate to the GDP of each member state, translates to an estimated 1700 million euros in Portugal. Consequently, this sector is primed for innovations in its practices, such as the traditional horseshoe application process.

Horses, depending on the demands placed on them, can walk either with or without horseshoes, with the vast majority using them. Unshod horses generally experience less impact damping when their hooves hit the ground compared to shod horses. This can be concerning not only for the hooves but also for the horse’s joints, as the shock transmitted to the limbs can reduce peak performance and overall longevity [3]. Therefore, the horseshoeing process is of the utmost importance to ensure the protection of the hooves as well as the horse’s performance [4].

Since medieval times, the horseshoe application process has followed a similar procedure: the horseshoe is heated until incandescent and then placed on the horse’s hoof to check its fit. It is then shaped while still hot until it properly fits the hoof. Next, the horse nails are hammered in, and once they emerge on the other side of the hoof, they are cut and bent against the hoof. Finally, the tips are smoothed with sandpaper to refine the hoof. During horseshoe replacement, the horseshoe is first removed by sanding the bent edges of the nails, allowing the horseshoe to be pulled off. These horseshoes are typically made from St or Al alloys [5]. As mentioned earlier, this is an ancient process, and the final quality is highly dependent on the skill of the operator.

Therefore, the process is open to modernization, particularly in the joining method, in which the aim would be to avoid the introduction of horse nails in the hoof while improving reproducibility and reducing the level of skill needed.

Adhesives, when compared to mechanical connections, have much more significant damping and shock-absorbing properties [6,7,8]. Additionally, they enable the use of other horseshoe materials that would be damaged by horse nails, such as polymers or composites, improving the shock-absorbing properties of the horseshoe even further. As said, even the more traditional metallic horseshoes have better damping characteristics than an unshod horse. However, the horseshoe material can also have a very significant influence on the injuries experienced by horses. According to Sprik et al. [5], who performed drop-weight tests in the same conditions at 8 m/s, determined the probability of limb bones breaking with a steel horseshoe as 75%, and for aluminum as 81%, while when using polyurethane (PU), the bones do not break. Lee and Choi [9] also demonstrated that CFRP horseshoes allow for better walkability and reduced fatigue in horses.

For that reason, adhesive-based solutions have been proposed to attach the horseshoe to the horse’s hoof [10,11,12]. However, these rely on the application of a paste adhesive, which can flow out of the bonded area, causing material waste and also requiring a second step for the removal of the excess adhesive. Furthermore, these adhesives always require a significant time to achieve full strength, which limits the mobility of the horse during this period. Additionally, the dismantling process still consists of simply pulling off the horseshoe by inserting a wedge in an indentation between the adhesive and the hoof [13] which has the potential to damage the hoof if not carefully performed. Although some adhesively bonded solutions have reached the market, their use is still not widespread because they do not instill the necessary confidence in either the farriers, riders, or owners of the horses.

On the other hand, one of the biggest drawbacks, if not the main drawback, associated with the use of adhesive joints to bond horseshoes at this point is still the fact that the currently used process is strongly rooted in the equestrian sector, which is highly averse to change. Therefore, very extensive suitability and durability studies need to be conducted to gain market trust. From research and interviewing horseshoe users, it is also clear that the reason behind these solutions not being widely implemented is that the adhesive interferes with the breathing and elasticity of the hoof [1].

This work aims to lay the groundwork for exploring alternative adhesive solutions to overcome the limitations of current methods. The initial step involved benchmarking and identifying the adhesive solutions currently used in glue-on horseshoeing and then determining their mechanical properties, which will serve as a reference for the development of future solutions. Consequently, two commercial acrylic adhesives, designated as Adhesive A and Adhesive B, were selected. The mechanical properties of both adhesives were assessed through a series of standardized tests, which were then utilized in numerical simulations to evaluate the predictability of in-joint behavior.

## 2. Experimental Details

### 2.1. Materials

In the present study, 2 acrylic commercially available adhesives developed for horseshoeing were characterized to establish a reference of mechanical properties useful for the development of future horseshoeing solutions. These two-component acrylic adhesives will be designated as Adhesive A and Adhesive B for confidentiality purposes.

Similar single-lap joints (SLJs) were manufactured with aluminum (Al 6082-T6, $\sigma_{y}$ = 255 MPa), high-strength steel (DIN 55 Si 7, $\sigma_{y}$ = 1100 MPa), designated as HSS throughout the paper, and mild steel (DIN St 33, $\sigma_{y}$ = 120 MPa) adherends.

These materials were selected because both Al and St are commonly used for the production of horseshoes. Usually, Al is used for high-performance horseshoes while mild St is used for standard ones [14].

Dissimilar SLJs were manufactured with mild St and horse hoof wall adherends. These joints are considered representative joints of the real one, which corresponds to the bonding of the horseshoe to the horse hoof. The authors of this paper are aware that the horse hoof wall specimens used in these joints do not fully replicate the real behavior of the surface to which the horseshoes are bonded nor the mechanical properties of the entire structure of the hoof. However, these specimens were chosen because they were the closest available material to realistically mimic the actual surface and behavior of the horse hoof.

A hoof from an adult horse (*Equus caballus* L.) with an unknown physical condition (destroyed for reasons outside the scope of this study) was obtained from the veterinarian hospital of Universidade de Trás-os-Montes e Alto Douro (Vila Real, Portugal), which removed all its living tissues and separated the hoof capsule from the underlying bone. Therefore, the provided sample consisted solely of the keratinized hoof material, without any associated soft tissue or bone.

### 2.2. Specimen Manufacturing

#### 2.2.1. Bulk Specimens

The bulk specimens were manufactured according to ASTM D412 [15] (injection molding fabrication), which led to 3 mm-thickness specimens [16]. Before use, the mold was degreased with acetone and coated with a release agent. The adhesive was poured into the assembled mold, and the hydrostatic pressure was ensured with 4 screws. The specimens were kept at room temperature (20 °C) for at least 7 days before testing. The dimensions of the bulk specimen are presented in Figure 1. Finally, the specimens were prepared for DIC analysis by coating them with white matte paint followed by the application of black speckling.

#### 2.2.2. TAST Specimens

A dedicated mold was used to manufacture 6 TAST specimens with a 0.5–0.8 mm-thickness adhesive layer and an overlap of 5 mm, which was ensured by 1.5 mm-thick shim tabs.

The adhesive thickness was measured for each specimen using a micrometer. Figure 2 displays the dimensions [mm] of the specimens following ISO 11003-2 standards [17]. The mold base was cleaned with acetone prior to the release agent application.

Firstly, the adhesive was applied to the lower adherends, followed by the lower shims (1.5 mm thickness) and the top adherends. Next, the upper shims were positioned on each specimen to prevent adhesive overflow onto undesired surfaces, ensuring an almost pure shear condition during testing [16]. Finally, the top plate was placed over the assembly. The specimens were cured at room temperature (20 °C) for 7 days with weight on top of the mold.

After curing, the specimens were carefully removed from the mold, and any excess adhesive was trimmed away using a sharp cutter. The shim tabs were then gently removed from each joint to avoid introducing cracks or damaging the joint.

#### 2.2.3. DCB Specimens

The DCB adherends (DIN 40 CrMnMo 8-6-4 St, $\sigma_{y}$ = 800 MPa) were used according to ISO 25217 standard [18]. The DCB specimens’ dimensions [mm] are displayed in Figure 3, where *t* represents the thickness of the adhesive layer and *a* the initial crack length.

The specimens were initially cleaned with acetone and then sandblasted. The bottom adherends were placed into a clean mold equipped with guiding pins and pre-coated with a mold release agent. To achieve an adhesive thickness of 0.20 mm, calibrated tape was applied to both sides of each specimen. A 0.10 mm sharp razor blade, with two pieces of 0.05 mm calibrated tape on either side, was used to create the pre-crack. These components were cleaned and coated with a release agent to ensure easy removal after curing. Afterwards, two loading blocks were bonded to the adherends, as suggested by Figure 4, which made possible the introduction of high-precision inclinometers into the test setup. These were responsible for the measurement of the rotations of each substrate at the respective loading point during the test.

The adhesive was applied to the bottom and top adherends, which were then assembled, and finally, the mold was closed and loaded with weights to avoid air entrapment. The curing conditions followed the ones from the previous sections.

#### 2.2.4. Horse Hoof Wall Adherends

The supplied hoof provided material for only 3 specimens. Each portion of the hoof wall for each substrate was isolated using an electric band saw (Figure 5c). The length of the substrate was limited by the size of the hoof, which is why dissimilar joints do not have the same dimensions as similar ones. Therefore, a comparison between similar and dissimilar joints will not be addressed. The thickness of the hoof wall adherends was obtained by manual machining (Figure 5e) in order to carefully separate the stratum medium (SM) tissue of the hoof wall from the remaining regions. Therefore, only the SM tissue was used in this study.

Although the hoof wall material in direct contact with the ground is where the horseshoe is bonded in a real scenario, this material was not suitable for specimen preparation due to insufficient material, so it was discarded. The region of the hoof wall facing the coffin bone was also excluded, as it contains tubule-forming dermal papillae [19].

After obtaining the hoof wall adherends, they were cleaned with isopropyl alcohol before the SLJ manufacturing process.

#### 2.2.5. SLJ Specimens

After adherend preparation, the SLJs were manufactured in an Al mold, previously cleaned and coated with release agent.

The Al mold employed for similar SLJs is designed for standard SLJs with overlap lengths of 12.5 mm, 25 mm, and 50 mm. However, given that the manufactured dissimilar SLJs are smaller than these standard dimensions, it was necessary to fabricate a 3D-printed mold. This custom mold was attached to the Al mold using pinned connections and was designed to match the geometry of the dissimilar SLJs, thereby ensuring accurate final joint dimensions.

Following assembly, the lap shear test specimens were kept at room temperature for 7 days with weights placed on top of the Al mold. After curing, the rig was opened, the joints were removed, and the excess adhesive was removed from the overlap area. In similar SLJs, St alignment tabs of 25 × 25 mm were bonded to the ends of each adherend, and for dissimilar SLJs, 3D-printed alignment tabs of 12 × 12 mm were used instead. These alignment tabs were used to minimize load misalignment and peel effects during testing.

The adherend materials used in the manufacturing of similar and dissimilar SLJs included Al, HSS, mild St, and horse hoof wall (HW), and the following adherend combinations were tested: Al–Al, HSS–HSS, Mild St–Mild St, and Mild St–HW. All joints were manufactured with a 12.5 mm overlap and a 0.2 mm adhesive layer thickness ensured by calibrated tape.

Similar SLJs

Al6082-T6 alloy, HSS, and mild St adherends were used to manufacture the Al–Al, HSS–HSS, and Mild St–Mild St SLJs, respectively. Both St and Al adherends were sandblasted on the overlap region and then cleaned with acetone prior to the manufacturing of SLJs. The adherends’ elastic properties are discussed in Section 2.1. The geometry of the similar joints is illustrated in Figure 6.

Dissimilar SLJs

Each dissimilar joint was fabricated with a mild St substrate and an HW substrate. Figure 7 displays the geometry used for the manufacturing of these joints. The mild St adherends were also sandblasted on the overlap region and then cleaned with acetone before the SLJ manufacturing process. No surface treatment was applied to the HW substrate to better replicate the real situation, which corresponds to the bonding of the horseshoe to the horse hoof.

### 2.3. Testing Setup

All tests were performed under quasi-static conditions at room temperature (20 °C) in an INSTRON^®^ 3367 (Illinois Tool Works, Hopkinton, MA, USA) quasi-static machine with a load cell of 30 kN. The adopted quasi-static test speeds were 2.5 mm/min for bulk tensile testing, 0.4 mm/min for TAST, 0.2 mm/min for DCB fracture tests, and 1 mm/min for SLJ testing. For each test, at least three specimens were tested.

#### 2.3.1. Tensile Test

Load–displacement curves were recorded for each test until failure occurred. For each test, it was necessary to synchronize the load applied on the specimen, which was recorded by the quasi-static machine, with the displacement field of the specimen’s areas of interest.

For that reason, a digital camera (Canon EOS M5, Canon Inc., Tokyo, Japan) equipped with a Canon EF-M 18–55 mm F/3.5–5.6 lens was placed in front of the specimens, with the lens oriented perpendicular to the observation surface. A laser beam was employed to ensure precise perpendicular alignment between the camera and the specimen. In order to determine the displacement field, a digital image correlation (DIC) approach was implemented using GOM Correlate^®^ (2019) software.

After determining the load–displacement curves, the engineering stress was obtained by dividing the load values by the initial cross-section of the specimen, and the engineering strain was determined considering that the deformation would be the displacement field obtained from DIC divided by the original gauge length of the bulk specimen, $L_{0}$.

Consequently, a stress–strain curve was generated for each specimen, allowing the determination of Young’s modulus, tensile strength, and tensile strain at failure for each adhesive.

#### 2.3.2. Thick Adherend Shear Test (TAST)

TAST specimens were mounted in the universal testing machine by connecting the end of each adherend to a gripping system, which was then assembled in the machine with stiff steel pins. The St adherends used for each specimen were considerably thicker than the adhesive layer in order to avoid peel stresses commonly seen in SLJ testing due to thin adherends (usually 2–3 mm). Therefore, TAST specimens avoided these mixed-mode load configurations, which ensured a more accurate determination of the shear properties [20]. The recording of the strain field was performed using a clip-on extensometer (2630-100 series, Instron, Norwood, MA, USA).

The engineering stress field was calculated by dividing the load values recorded by the quasi-static machine by the overlap area of the specimens. Finally, a stress–strain curve was obtained for each specimen, followed by the determination of the shear properties of the adhesives: shear modulus and shear strength.

#### 2.3.3. Double Cantilever Beam Test

The DCB specimens were installed in the universal testing machine by connecting the loading blocks to the machine’s gripping system with pins.

The J-integral approach requires the synchronization of the load applied on the specimen throughout the test and the rotation angles at the load introduction points. Thus, the load–displacement curve was registered by the quasi-static machine and the rotation angles were recorded with Seika^®^ NA2-10 inclinometers (Wiggensbach, Germany). The DCB specimens were loaded in tension with a small pre-load applied to ensure a stable crack propagation.

Data-Reduction Scheme

Several data-reduction methods have been used in order to determine the fracture toughness of a material, such as the Simple Beam Theory (SBT), Corrected Beam Theory (CBT), Experimental Compliance Method (ECM), and Compliance-Based Beam Theory (CBBM). All these methods are based on Linear Elastic Fracture Mechanics (LEFM) [21] and can only be applied if the fracture process zone (FPZ) is very small when compared to the length of the component. In order to surpass the limitations of LEFM, other methods have been proposed based on the J-integral approach [22], which is a contour integral that accounts for the energy per unit area needed to create two new surfaces on a cracked body under loading conditions. Rice [23] demonstrated that the J-integral is independent of the contour for any elastic or elastoplastic material and therefore allows the presence of plasticity [24], which means that the calculation of the fracture energy will be more accurate, especially in the case of more ductile adhesives that involve large FPZs.

Preliminary tests indicated that both adhesives examined in this study exhibited high ductility, as will be discussed in Section 4.1.1. Consequently, the assumptions of LEFM were not applicable, and thus the J-integral approach was used to determine the Mode I fracture toughness for each adhesive.

The loads and rotations measured during the tests were applied to the J-integral approach to determine the fracture energy for each test. In a mode I loading configuration, the value of *J* along the DCB specimen’s exterior boundary is given by [25]:(1)$$J_{I\;}=\frac{P}{b}\;\left(\theta_{up}-\theta_{low}\right)$$
where *P* is the applied load, *b* is the width of the specimen, and $\theta_{up}$ and $\theta_{low}$ are the relative rotations of the upper and lower adherends at the loading points, respectively. This relation is applicable due to the path-independence property of the J-integral in both linear and nonlinear elastic materials while assuming that the region outside the fracture process zone (FPZ) remains elastic during the fracture process. This assumption is commonly accepted in fracture mechanics as it ensures the applicability of the J-integral to evaluate stress intensity factors and fracture toughness in the material [23].

#### 2.3.4. Single Lap Joints

Each joint was assembled on the gripping system of the machine by fixing the end of one adherend to the lower fixture of the machine, which remains static throughout the test, and fixing the end of the opposite adherend to the machine’s crosshead resorting to pins. The load–displacement curves were recorded by the machine’s software from the beginning of the test up until the specimen’s failure.

## 3. Numerical Modelling

ABAQUS^®^ FE 2017 software was used to perform a numerical analysis where cohesive zone modelling (CZM) was applied to the adhesive layer, resorting to traction-separation laws for mode I and mode II.

The St adherends used in the DCB test procedures were modeled following a linear elastic approach, as were the Al, HSS, and mild St adherends used in similar SLJs. This modelling approach was justified by the observation that the failure loads from the experimental tests indicated that these adherends did not exceed their yielding point, as will be discussed in Section 4.1.4.

All adherends were discretized using ABAQUS^®^ CPE4R plane-strain elements with reduced integration and hourglass control.

The adhesive layer was modelled using COH2D four-node cohesive elements, selected due to their compatibility with the elements chosen for the adherends. For SLJ modelling, a mesh refinement procedure was performed along the adhesive overlap length using a double-bias parameter (0.05–0.2 mm in the x-direction and 0.2–0.3 mm in the y-direction) as suggested by Figure 8, allowing for an increase in mesh density at the overlap edges due to high stress concentrations in these locations.

The chosen boundary conditions simulated the fixtures used in the experimental tests. The bottom edge of one of the adherends was fixed, and a displacement load was applied to the opposite edge of the other adherend, as displayed in Figure 9.

For the DCB model, mesh refinement was applied to the adherends in the y-direction towards the cohesive layer using a single-bias parameter (Figure 10). This approach increased the mesh density in these regions, which is essential for accurately capturing the stress distribution and modelling crack propagation effectively.

As for boundary conditions, the hole of the bottom adherend was pinned (Figure 11—RP1), allowing for rotation along the normal axis to the surface, and a displacement along the *y*-axis was applied to the hole of the upper substrate (Figure 11—RP2).

Cohesive Damage Model

For a pure mode loading (pure Mode I or II), the elastic behavior of the adhesive was described by a linear relationship between traction, σ, and separation, δ, before damage starts to grow. This relation is dependent on the tensile and shear stiffness parameters. These are defined as the ratio between the Young’s modulus (Mode I) or the shear modulus (Mode II) and the adhesive thickness (*t*). After the elastic region, the softening phenomenon begins, leading to a traction separation law defined by:(2)$$\sigma =\left(I-D\right)\delta$$
where *I* is the identity matrix and *D* is a diagonal matrix containing the damage parameter, *d*, in the position corresponding to mode I and mode II.

The objective is to define a damage law that can reproduce the behavior of the adhesive in its respective bonded joints. The triangular shape law is mathematically simple as it assumes a linear elastic response up to the peak stress, followed by progressive damage propagation after the peak stress. Its simplicity enhances computational efficiency, thus reducing the risk of convergence issues. For these reasons, it is a common and simple form for simulating adhesive behavior [26]. However, one of its limitations is the inability to capture the plastic deformation behavior that occurs before damage initiation in ductile adhesives, where stress may remain constant or increase at a reduced rate.

In shear-dominated loading (Mode II), ductile adhesives often show significant plastic deformation before failure. Therefore, the triangular law may not accurately predict the stress distribution or energy dissipation in such scenarios, leading to discrepancies between numerical and experimental results.

As a result, a triangular law was used to describe the adhesive’s performance in mode I due to its simplicity and its ability to properly represent the adhesive’s behavior in this configuration, as will be discussed in Section 4.2.1. Nevertheless, both adhesives exhibited a nonlinear stress–strain relationship in pure shear conditions due to plastic deformation before the peak stress, as will be discussed in detail in Section 4.1.2. Consequently, a triangular law will not be able to capture this behavior. Pinto et al. [27] proposed a cohesive damage model, characterized by a trapezoidal shape with an increasing stress plateau, to simulate the behavior of a ductile adhesive in a SLJ configuration. Given the similarity in mechanical properties between the adhesives studied in this paper and the adhesive analyzed in [27], the same trapezoidal law was adopted to characterize the adhesive’s behavior in mode II.

For mode I, the damage parameter following a triangular shape law is expressed as follows:(3)ⅆI=δIfδI−δI0δIδIf−δI0
where δI0 represents the displacement at damage initiation and δIf represents the displacement at complete failure [27].

For mode II, the damage parameter following the trapezoidal shape of Figure 12 is expressed as follows:(4)$$d_{II}=\frac{{(\delta }_{II}-\delta_{1})(\delta_{2}\;\alpha_{II}-\delta_{1})}{\delta_{II}\;\left(\delta_{2}-\delta_{1}\right)}\;\;\;\;\;\delta_{1}\leq \delta_{II}\leq \delta_{2}$$
(5)$$d_{II}=1+\frac{\alpha_{II}\delta_{1}\left(\delta_{II}-\delta_{u}\right)}{\delta_{II}\left(\delta_{u}-\delta_{2}\right)}\;\;\;\;\;\;\;\;\;\;\;\;\delta_{2}\leq \delta_{II}\leq \delta_{u}$$
where $\alpha_{II}$ stands for the strength ratio
(6)$$\alpha_{II}=\frac{\sigma_{u}}{\sigma_{s}}$$
where $\sigma_{u}$ corresponds to the ultimate strength and $\sigma_{s}$ corresponds to the softening strength in mode II [27].

The maximum displacement, $\delta_{u}$, at which complete failure occurs, is determined by equating the area under the softening curve to the critical fracture energy in mode II ($J_{IIc}$):(7)$$J_{IIc}=\frac{\sigma_{s}}{2}\left(\delta_{2}+\alpha_{II}\left(\delta_{u}-\delta_{1}\right)\right)$$

Usually, bonded joints are under mixed-mode loading and therefore a mixed-mode damage model had to be implemented. The selected damage criterion was a quadratic nominal stress formulation, which is described by
(8)$${\left(\frac{\sigma_{I}}{\sigma_{max}}\right)}^{2}+{\left(\frac{\sigma_{II}}{\sigma_{s}}\right)}^{2}=1$$
where $\sigma_{max}$ represents the maximum strength in mode I. Therefore, damage begins as soon as this criterion is satisfied.

The definition of the damage criterion for traction–separation laws is based on an energetic approach, where the critical energy release rate values for both mode I and mode II are considered. Thus, it is possible to extrapolate a mixed-mode fracture energy relationship by combining the individual laws into a unified mixed-law formulation. In this case, a linear energetic criterion was used:(9)$$\frac{J_{I}}{J_{Ic}}+\frac{J_{II}}{J_{IIc}}=1$$

Cohesive Parameters

The cohesive damage model was implemented in the adhesive layer (*t* = 0.2 mm) of the adhesive in pure modes I and II. The cohesive parameters in pure mode I to be defined are the maximum strength ($\sigma_{max}$), the elastic stiffness (*k_I_*), and the critical fracture energy ($J_{Ic}$), and in pure mode II are the maximum strength ($\sigma_{u}$), the softening strength ($\sigma_{s}$), the second inflexion point ($\delta_{2}$), and the critical fracture energy ($J_{IIc}$). For pure mode I, these parameters were assumed to be equal to the bulk quantities, while $J_{Ic}$ was determined from DCB tests. For pure mode II, the maximum and softening strengths were approximately the same as the TAST quantities as well as the shear modulus. The first inflexion point ($\delta_{1}$) was calculated from the ratio between the softening strength and the initial shear stiffness of the adhesive ($k_{I}$) and the second inflexion point ($\delta_{2}$) was adjusted to an optimal parameter.

Since fracture tests were only performed for mode I, only the critical fracture energy in mode I was determined, and therefore the critical fracture energy in mode II was estimated according to the literature [27].

## 4. Results and Discussion

### 4.1. Experimental Tests

#### 4.1.1. Tensile Test

The surface of the dog-bone specimens was sprayed with matte white paint prior to speckling with dark matte paint. For each adhesive, three specimens were positioned in pinned fixtures for testing. The strains were computed using DIC software, and the stress–strain curve was obtained. The stress–strain curves are illustrated in Figure 13. Table 1 summarizes the average values as well as the standard deviation for Young’s modulus, tensile strength, and tensile strain to failure for Adhesive A and Adhesive B.

Based on these results, the Poisson’s ratio, ν, was estimated using DIC analysis, yielding values of ν = 0.30 ± 0.01 and ν = 0.34 ± 0.03 for Adhesive A and Adhesive B, respectively.

Both adhesives display similar tensile properties, suggesting comparable formulations. As shown in Figure 13, the adhesives exhibit highly ductile behavior, with strain-to-failure values reaching approximately 85% for Adhesive A and 65% for Adhesive B.

The area under the stress–strain curve suggests that the material has a high energy absorption capacity, which is beneficial in impact situations such as walking, trotting, and galloping activities. It is also advantageous for accommodating the natural expansion, contraction, and flexing of the horse’s hoof.

#### 4.1.2. Thick Adherend Shear Test

The thick adherend shear test strain field was obtained using extensometer-based displacement measurements. All specimens showed a similar failure mode, as illustrated in Figure 14, revealing cohesive failure, which indicated a good adhesion between the adhesive and the adherends. The stress–strain curves are displayed in Figure 15. The shear modulus was calculated using the curve’s slope in the elastic region. These results are summarized in Table 2.

According to Table 2, the shear modulus obtained from the TAST is 211 ± 17 MPa and 235 ± 63 MPa for Adhesives A and B, respectively. Considering the Poisson’s ratio relation between the Young’s and shear modulus in isotropic materials:(10)$$E=2G\left(1+\nu \right)$$

The calculated Poisson’s ratio would be 0.35 and 0.36 for Adhesives A and B, respectively, which are close enough to the values determined using the DIC method (0.30 for Adhesive A and 0.34 for Adhesive B).

The tensile strength of each adhesive is higher than its shear strength, as shown in Table 1 and Table 2, and these values are relatively close due to the high ductility of these adhesives.

For example, when the horse moves, there can be forces that try to pull the horseshoe away from the hoof (e.g., due to the horse’s weight or impact on the ground). In this case, tensile forces are predominant, and a higher tensile strength will be able to resist this pulling force and hold the horseshoe in place without detaching.

In the case of shear forces, these occur when there is a sliding motion, such as when the horse walks or runs and the hoof slides along the ground or when the foot is in motion. Therefore, good shear strength is also needed in order to prevent the horseshoe from sliding or detaching from the hoof during the horse’s movement.

Consequently, this balance between shear and tensile strength might be beneficial, depending on the specific forces the horse will experience during activities.

#### 4.1.3. DCB Test

For each adhesive, three DCB specimens were used to obtain the critical energy release rate, $G_{Ic}$. The average curve and its corresponding standard deviation of the DCB test results for both adhesives can be observed in Figure 16. The load peak at which the crack propagation initiates is similar to the studied adhesives, with Adhesive B exhibiting a 10% higher peak load compared to Adhesive A. After the peak load is reached, the crack propagation behaves similarly for both adhesives, as depicted in Figure 16. Figure 17 displays the typical fracture surface for the quasi-static tests for Adhesive A and Adhesive B. The two fracture surfaces show a cohesive fracture behavior, which corroborates the good adhesion between the adhesive and the steel adherends as noted in the TAST results, ensuring that the fracture energy obtained belongs to the adhesive and not to the interface.

After testing the DCB specimens, the J-integral data-reduction method was used to assess the fracture energy of each adhesive in quasi-static conditions, resorting to Equation (1). The average curve with the corresponding standard deviation is presented in Figure 18.

The J vs. displacement curves present identical behavior, with a parabolic part at the beginning of the tests followed by an almost linear part that corresponds to the crack propagation. At this stage, it is safe to consider that there is damage in the adhesive layer, and with the damage increase, the adhesives are gradually losing their cohesive capacity. Fracture toughness of $G_{Ic}$ = 1.3 ± 0.07 N/mm and $G_{Ic}$ = 1.24 ± 0.15 N/mm was found for Adhesives A and B, respectively, which once again supports the similar formulations of the adhesives.

These fracture toughness values are characteristic of more ductile adhesives, allowing them to absorb greater amounts of energy before failure and providing enhanced resilience against stress or impact compared to brittle adhesives.

As previously mentioned, the fracture toughness in mode II was estimated according to literature values [27] that consider an adhesive with mechanical properties identical to those of the adhesives studied in this work. Thus, a fracture toughness in mode II of $G_{IIc}$ = 6.0 N/mm was considered as a preliminary value for the numerical simulation.

Table 3 provides an overview of the tensile, shear, and fracture properties determined for each adhesive.

#### 4.1.4. Similar SLJ

As shown in Table 3, the mechanical properties of each adhesive are almost identical, and therefore it was decided to only present the results of the similar SLJs of Adhesive A. Preliminary SLJ tests were also conducted using Adhesive B to verify that the lap shear strength of both adhesives was comparable, and it was concluded that the in-joint behavior of both adhesives was also similar.

Before testing, St alignment tabs of 25 × 25 mm were bonded to the end of each adherend to reduce load misalignment and peel effect when testing.

Figure 19 shows the failure mode of Al–Al SLJs, HSS–HSS SLJs, and Mild St–Mild St SLJs for Adhesive A, revealing cohesive failure. This indicates strong adhesion between the sandblasted Al adherends and the adhesive, as well as between the sandblasted St adherends and the adhesive. This enabled the assessment of the adhesive’s lap shear strength rather than that of the interface.

In Figure 20, the average curve and its standard deviation of the load–displacement curves and lap shear stress evolution are shown for Adhesive A, considering every studied adherend combination. The shaded area in each curve represents the load standard deviation and the error bar shows the displacement standard deviation.

Based on Figure 20, the maximum lap shear strength for the Al–Al SLJ is 16.0 ± 0.3 MPa, for the HSS–HSS SLJ is 16.5 ± 0.4 MPa, and for the Mild St–Mild St is 16.1 ± 0.3 MPa for Adhesive A. As the adherends remained in the elastic region and did not exhibit plastic deformation prior to adhesive failure, the behavior of the joint is mainly governed by the adhesive’s properties. Consequently, the change in slope between 3 and 4 kN (Figure 20) corresponds to the yielding of the adhesive.

However, the change in slope observed in the Al–Al SLJ was less pronounced compared to the St–St combinations. This behavior can be attributed to the significantly lower modulus of elasticity of Al (approximately 1/3 that of St). Therefore, after 4 kN, Al actively contributes to the deformation of the joint together with the adhesive. Since St is much stiffer than Al, at this load level, the deformation of the joint will be governed by the adhesive.

The failure load was similar for every adherend combination, reaching approximately 5 kN. However, as expected, the displacements at the failure of the Al–Al SLJs were higher than those of St–St SLJs due to lower stiffness of the Al adherends when compared to those of St.

Therefore, the lap shear strength values of the studied adhesives should serve as a reference for any future adhesive solutions developed for the horseshoeing process.

This demonstrates that the performance of the adhesive is consistent for Al and St adherends, with respect to the maximum lap shear strength. This is promising, particularly in the context of the horseshoeing process, where both Al and St horseshoes are commonly used. Consequently, it can be ensured that the adhesive will exhibit similar performance for either material as its adhesion properties remain effective regardless of the horseshoe material.

#### 4.1.5. Dissimilar SLJ

As mentioned earlier, the dissimilar SLJs were manufactured with mild St and horse hoof wall adherends. Before SLJ testing, 3D-printed alignment tabs of 12 × 12 mm were bonded to the end of each adherend to avoid load misalignment and peel effects, following the same procedure as for similar joints. Three specimens of dissimilar SLJs were tested, and their load–displacement curves are displayed in Figure 21.

As depicted in Figure 21, all specimens exhibited identical performance, with failure consistently occurring in the hoof substrate, as seen in Figure 22. However, the joints labeled “test1” and “test3” displayed notable sudden load drops followed by immediate recovery. These fluctuations may be attributed to defects introduced in the hoof adherends during the extraction process from the molds and the subsequent cleaning prior to testing.

During the cleaning process of joint “test3”, residues of 3D-printed material from the mold remained adhered to the hoof substrate, particularly near the overlap area. To avoid causing additional damage to the same adherend, it was decided to proceed with testing of the joint in this condition (Figure 23). Consequently, the presence of these 3D-printed material traces may have contributed to a higher peak load compared to the other tests.

The failure of all dissimilar joints within the hoof substrate indicates that, at least under the current loading conditions, these commercial adhesives possess greater strength than the hoof material. Therefore, if a real joint were subjected to identical loading conditions, it is likely that the horseshoe would detach from the hoof, potentially carrying away hoof material. However, this conclusion cannot be definitively stated, as the hoof substrate used for testing the dissimilar joints does not precisely replicate the mechanical properties or surface roughness of the actual region where the horseshoe is attached, as it originates from the SM region of the hoof wall.

### 4.2. Numerical Validation

Given the similar mechanical properties of the two adhesives investigated, only Adhesive A will be considered for the development and validation of the numerical model.

#### 4.2.1. Mode I

The cohesive law of the adhesive in mode I was modeled with a triangular shape, characterized by an initial elastic region, followed by damage initiation, which progressively degrades the adhesive’s properties. To validate the considered cohesive law, a DCB specimen was numerically simulated during its opening movement, which corresponds to pure mode I loading. The resulting numerical load–displacement curve was then compared to experimental data to assess the agreement between the numerical and experimental results (Figure 24).

Figure 24 demonstrates a strong correlation between the numerical and experimental results, which indicates that a triangular traction–separation law is sufficient to characterize the adhesive’s performance in pure mode I. The initial stiffness of the specimen was well captured by the numerical model, and the numerical peak load was also within the standard deviation of the experimental data. The initial crack propagation occurring after 1 mm displacement was also similar between the numerical and experimental curves.

#### 4.2.2. Similar SLJ

The cohesive law of the adhesive in mode II was modeled with a trapezoidal shape with increasing stresses, as detailed in Section 3. To validate this cohesive law, a SLJ configuration was modeled using the considered similar adherend combinations, i.e., Al–Al and St–St. The SLJ test does not fully correspond to pure mode II loading but rather to a mixed-mode loading condition. However, the SLJ configuration was selected for mode II validation due to the dominance of in-plane shear stresses throughout the test. To further support this approach, various shapes of cohesive laws in mode I were considered for the SLJ configuration to examine their effect on the load–displacement response. As anticipated, the influence of the mode I cohesive law shape on the P–δ curve was minimal, thereby confirming the suitability of the SLJ configuration for validating the mode II cohesive law.

Figure 25, Figure 26 and Figure 27 show the comparison between the experimental and numerical load–displacement curves for the Al–Al, HSS–HSS, and Mild St–Mild St SLJs of Adhesive A.

The cohesive laws described in Section 3 were applied, resulting in the numerical curves shown in Figure 25, Figure 26 and Figure 27. A comparison between the experimental data and the numerical curves for each adherend combination reveals that the numerical model accurately predicted the failure load of all joints, which was approximately 5 kN. Additionally, the stiffness of each joint was well captured, as the initial elastic region of the numerical curves closely matched the experimental data.

However, discrepancies were observed in the displacement at failure, with the numerical displacements consistently exceeding the experimental values. Specifically, for the Al–Al SLJs, the numerical displacement at failure was 45% higher than the experimental value, while for the St–St SLJs, it was approximately 60% higher.

These discrepancies in failure displacement may be attributed to the strain rate-dependent behavior of ductile adhesives. As noted earlier, the adhesive included in the numerical model has ductile behavior and it is therefore strain rate-dependent. The adhesive’s mechanical properties were determined at crosshead speeds recommended by the relevant standards, which do not match the strain rate used in the simulations. This mismatch may explain the observed differences in displacement at failure.

The numerical model also reveals that as soon as the entire adhesive layer begins to experience damage (t = 0.108 s as shown in Figure 28), a change in slope occurs in the pdelta curve (Figure 27), which indicates the onset of plastic deformation in the adhesive.

As anticipated, the stresses perpendicular to the loading direction are nonzero (Figure 29), confirming the presence of mixed-mode loading during the SLJ test. These stresses contribute to the peel effects experienced by the joint. Nonetheless, it is clear that shear stresses (11) are dominant as they are approximately 12.5 times higher than the normal stresses (22).

Despite the limitations of the numerical model, it provided a reasonable prediction of joint behavior, particularly in terms of failure loads.

With the validation of the cohesive laws in mode I and mode II, it is now possible to simulate more complex loading conditions involving significant mixed-mode loading, as encountered in the real joint configuration, and also more complex geometries.

## 5. Conclusions

This study examines the mechanical properties of two commercial adhesives currently used in the equine industry, specifically for horseshoeing applications. Both adhesives are acrylic-based, providing an optimal balance of elastic and ductile properties along with the necessary strength for this purpose.

A series of standardized tests revealed that the mechanical properties of both adhesives are quite similar, suggesting a comparable formulation. Therefore, the in-joint behavior in similar SLJs was only analyzed for Adhesive A, resulting in a lap shear strength of approximately 16 MPa for every adherend combination.

This value may serve as a benchmark for developing innovative bonding solutions that could surpass the limitations of the adhesives studied here.

A numerical model was successfully implemented using a cohesive zone modelling (CZM) approach to simulate the performance of the tested similar joints. A triangular traction–separation law was applied for mode I, while a trapezoidal law with increasing stresses was utilized for mode II. These cohesive laws allowed for a moderate prediction of the behavior of similar joints.

The next phase of this research is the development of a model of the real joint by incorporating the geometry of both the hoof and horseshoe into the assembly. This model will then simulate the joint’s behavior under realistic geometrical and loading conditions. This approach aims to identify regions requiring adhesive reinforcement, as well as areas where adhesive use may be minimized.

## Figures and Tables

**Figure 1 biomimetics-10-00002-f001:**
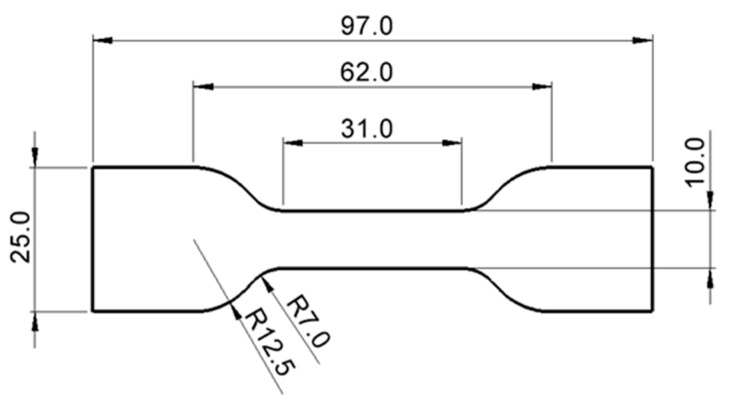
Geometry of the specimens for tensile testing of the adhesives [mm] according to ASTM D412.

**Figure 2 biomimetics-10-00002-f002:**
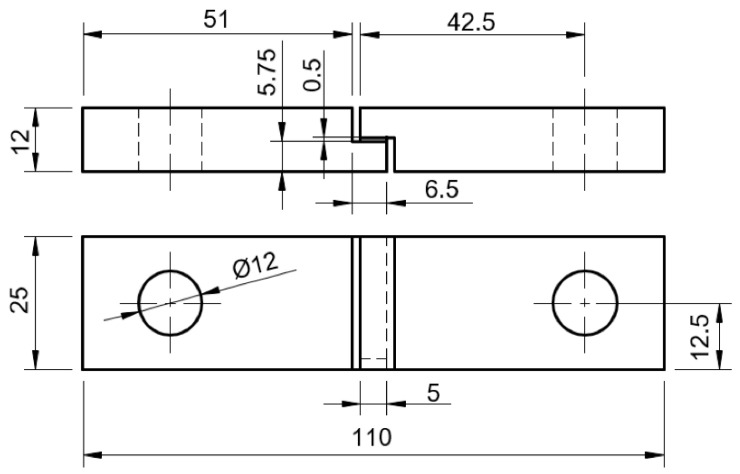
Geometry and dimensions of TAST specimens [mm] according to ISO 11003-2.

**Figure 3 biomimetics-10-00002-f003:**
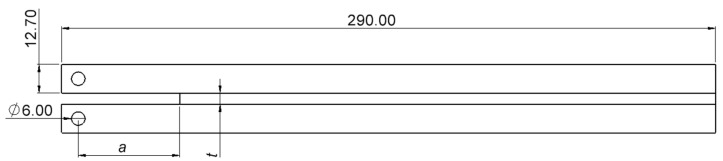
DCB specimen geometry and dimensions [mm].

**Figure 4 biomimetics-10-00002-f004:**
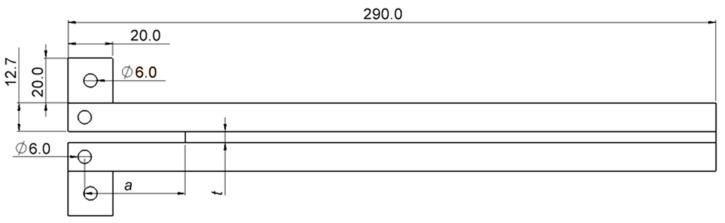
DCB specimen geometry with loading blocks for the introduction of inclinometers in the testing setup.

**Figure 5 biomimetics-10-00002-f005:**
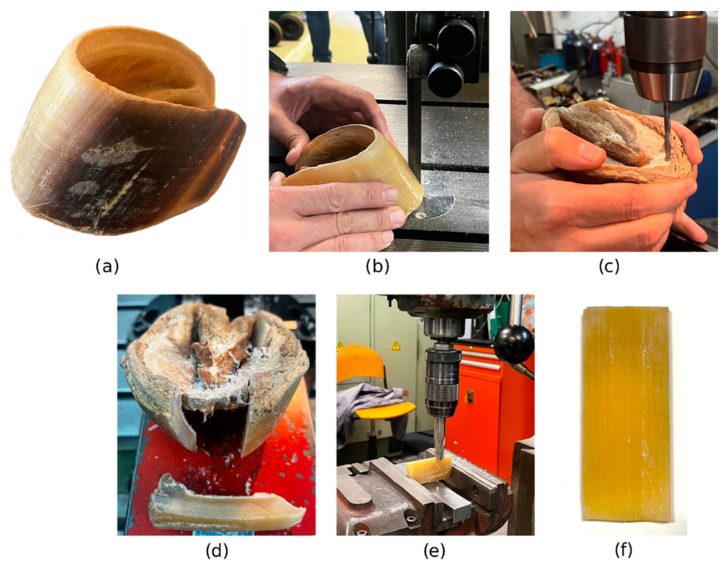
Manufacturing process of the hoof wall adherends from the initial and natural hoof wall (**a**) to the final substrate (**f**). The process involved cutting the hoof wall with a thin electric band saw (**b**) to an intended width, then inducing holes in the sole (**c**) until extraction of the specimen (**d**). The final step is the milling of the specimens (**e**) in order to reach the required dimensions (**f**).

**Figure 6 biomimetics-10-00002-f006:**
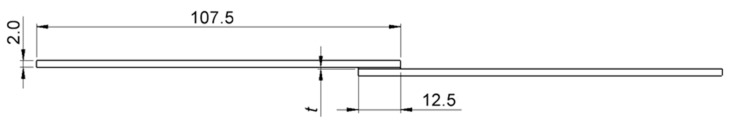
Similar SLJ geometry. The width (out of plane) of each adherend is 25 mm, and *t* represents the thickness of the adhesive layer, which was defined as 0.2 mm.

**Figure 7 biomimetics-10-00002-f007:**
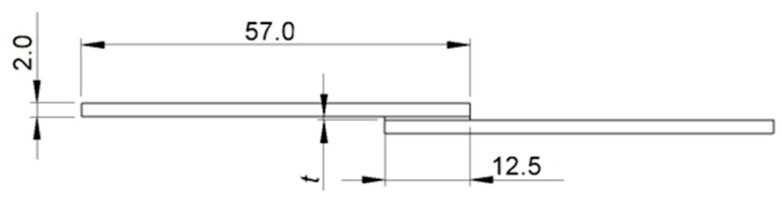
Dissimilar SLJ geometry. The width of each adherend is 12.5 mm, and *t* represents the thickness of the adhesive layer.

**Figure 8 biomimetics-10-00002-f008:**
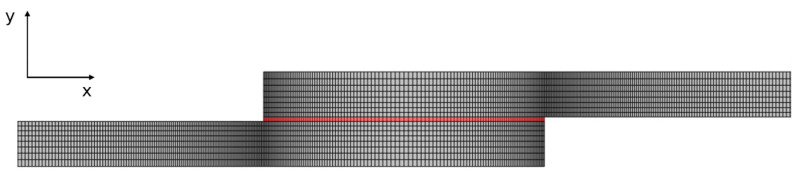
Details of the mesh used for similar SLJs at the overlap region.

**Figure 9 biomimetics-10-00002-f009:**
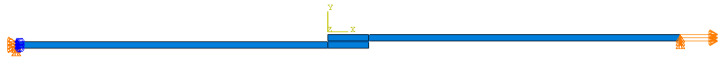
Boundary conditions of the SLJ model.

**Figure 10 biomimetics-10-00002-f010:**
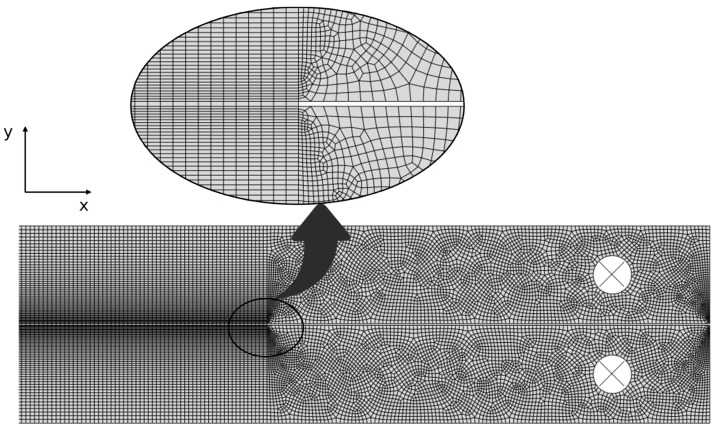
Details of the mesh used for simulating DCB mode I at the cohesive layer.

**Figure 11 biomimetics-10-00002-f011:**
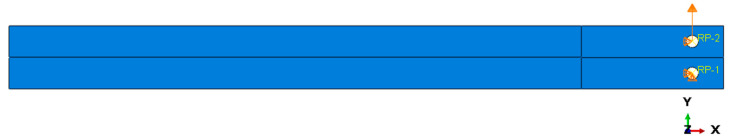
Boundary conditions for the DCB model.

**Figure 12 biomimetics-10-00002-f012:**
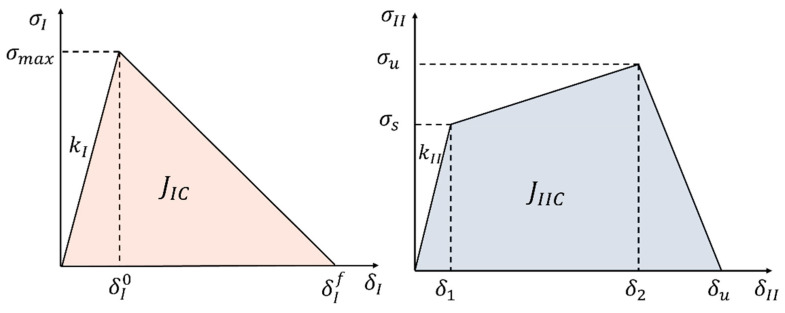
Triangular and trapezoidal shape cohesive laws used for characterizing the behavior of the adhesive in mode I and II, respectively.

**Figure 13 biomimetics-10-00002-f013:**
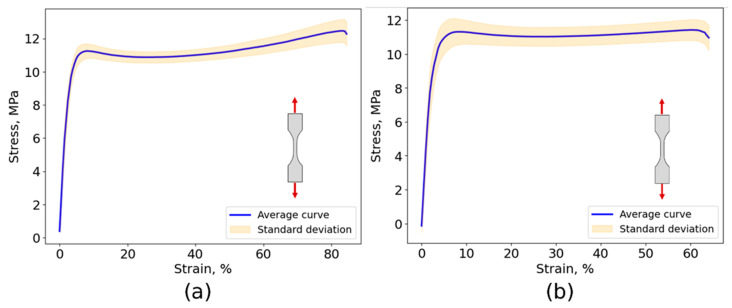
Tensile stress–strain curve of (**a**) Adhesive A and (**b**) Adhesive B.

**Figure 14 biomimetics-10-00002-f014:**
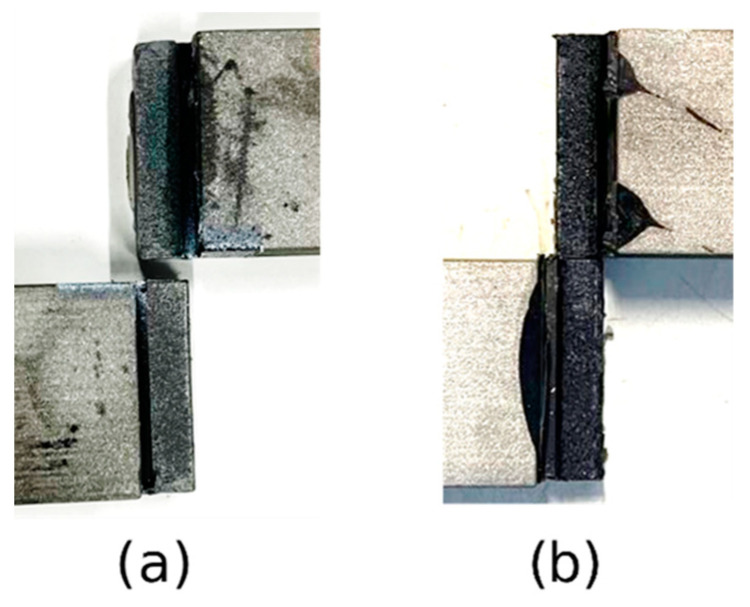
Failure mode of TAST specimens: (**a**) Adhesive A, (**b**) Adhesive B.

**Figure 15 biomimetics-10-00002-f015:**
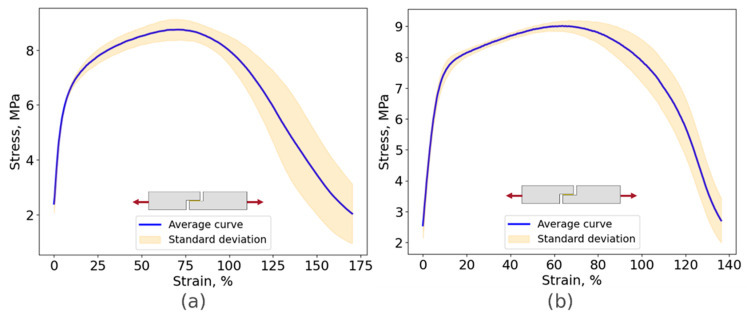
Stress–strain curve for the TAST specimens: (**a**) Adhesive A, (**b**) Adhesive B.

**Figure 16 biomimetics-10-00002-f016:**
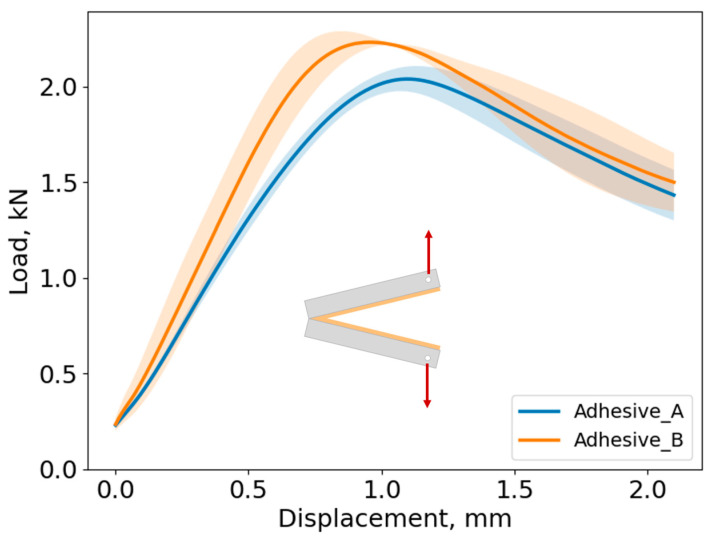
Load–displacement curves of the studied adhesives, where the full line represents the average curve of the shaded area represented the standard deviation.

**Figure 17 biomimetics-10-00002-f017:**
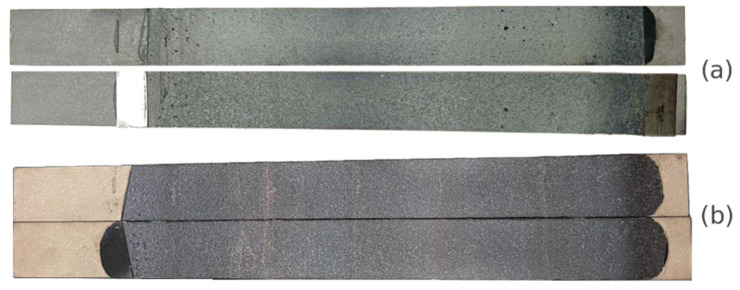
Failure mode of the DCB specimens for (**a**) Adhesive A and (**b**) Adhesive B, showing cohesive failure for both adhesives.

**Figure 18 biomimetics-10-00002-f018:**
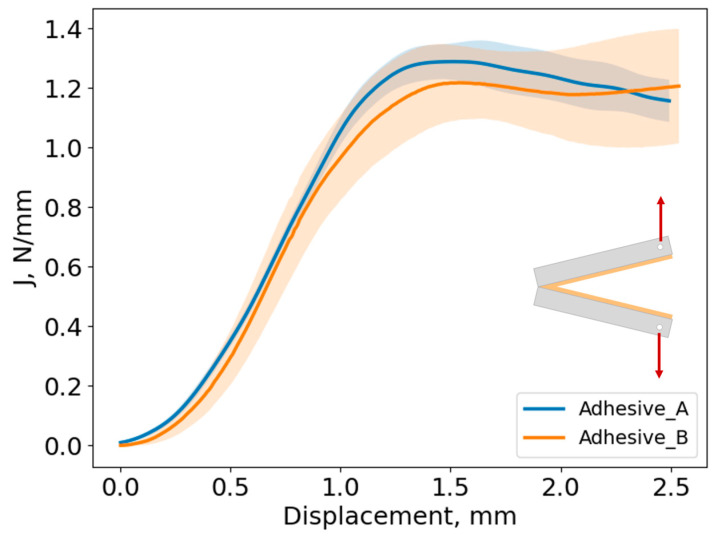
J vs. loading point displacement curves for Adhesives A and B, where full line represents the average curve and the shaded area represents the standard deviation.

**Figure 19 biomimetics-10-00002-f019:**
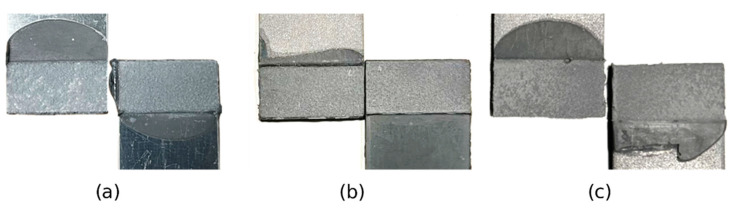
Failure mode of the similar SLJ: (**a**) Al–Al SLJs, (**b**) HSS–HSS SLJs, and (**c**) Mild St–Mild St SLJs, showing cohesive failure in every adherend combination.

**Figure 20 biomimetics-10-00002-f020:**
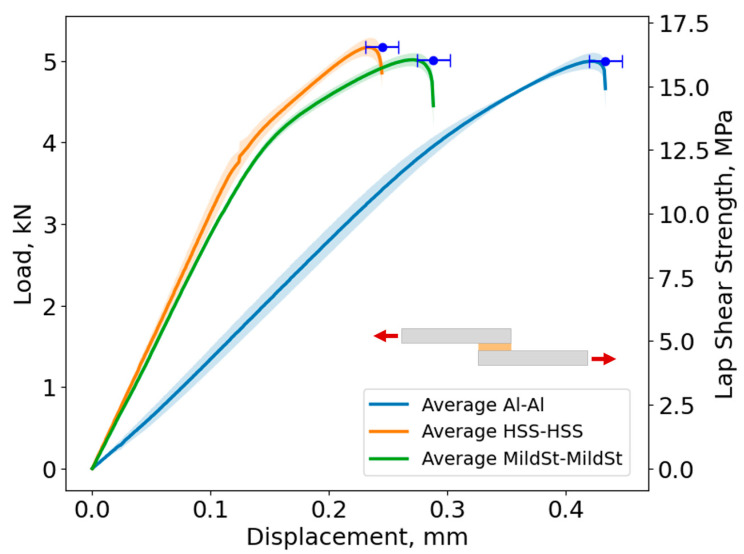
Load–displacement curves of similar SLJs (Adhesive A) for every adherend combination, where the full line represents the average curve and the shaded area the standard deviation.

**Figure 21 biomimetics-10-00002-f021:**
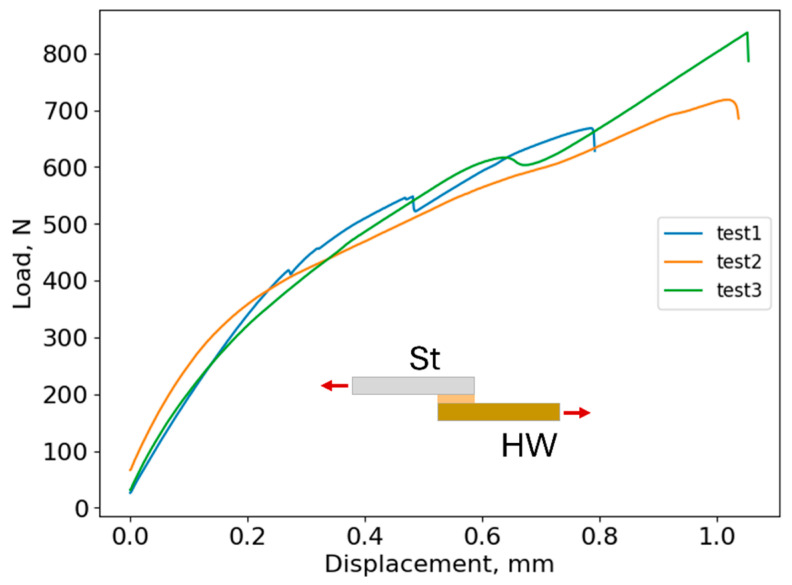
Load–displacement curves of the tested dissimilar joints.

**Figure 22 biomimetics-10-00002-f022:**
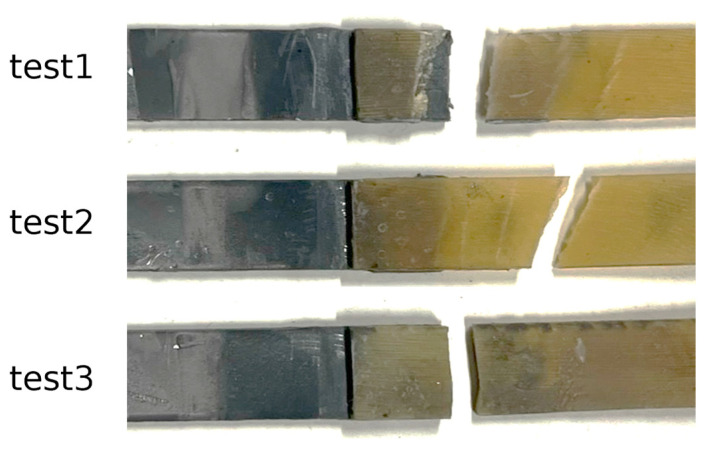
Failure mode of the tested dissimilar joints, showing a substrate failure in every test.

**Figure 23 biomimetics-10-00002-f023:**
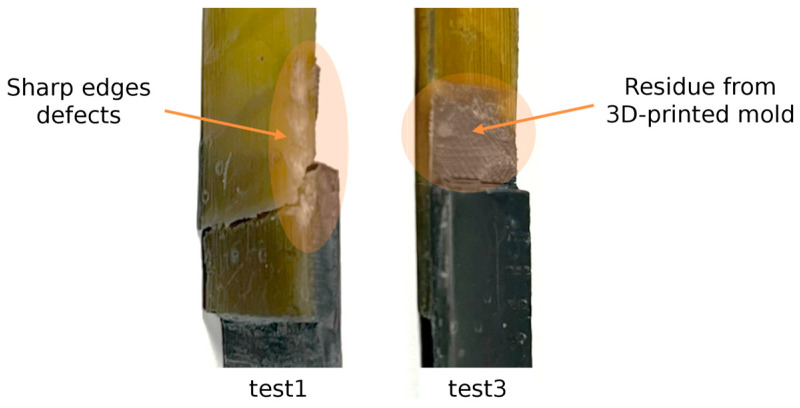
Joint defects created during the cleaning process.

**Figure 24 biomimetics-10-00002-f024:**
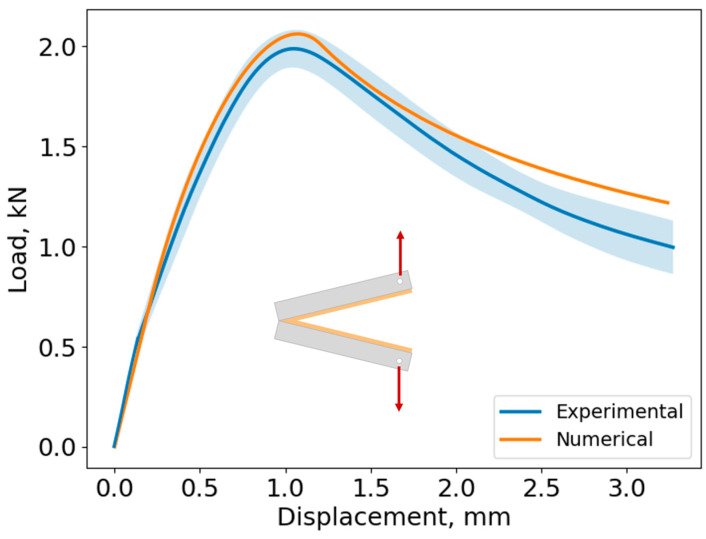
Load vs. displacement curve for both numerical and experimental results of DCB specimens (Mode I). The shaded blue area represents the standard deviation of the experimental results.

**Figure 25 biomimetics-10-00002-f025:**
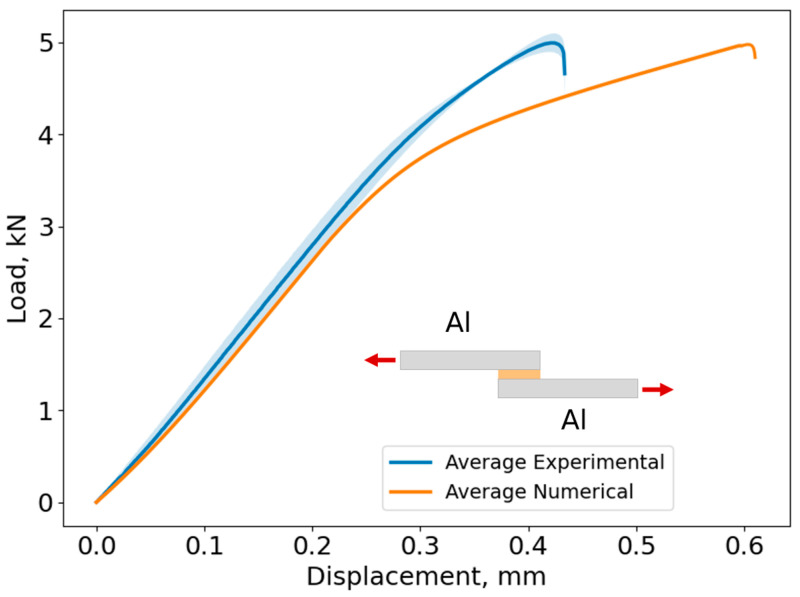
Load–displacement curves for numerical and experimental results of Al–Al SLJs of Adhesive A. The shaded area in the average experimental curve corresponds to standard deviation.

**Figure 26 biomimetics-10-00002-f026:**
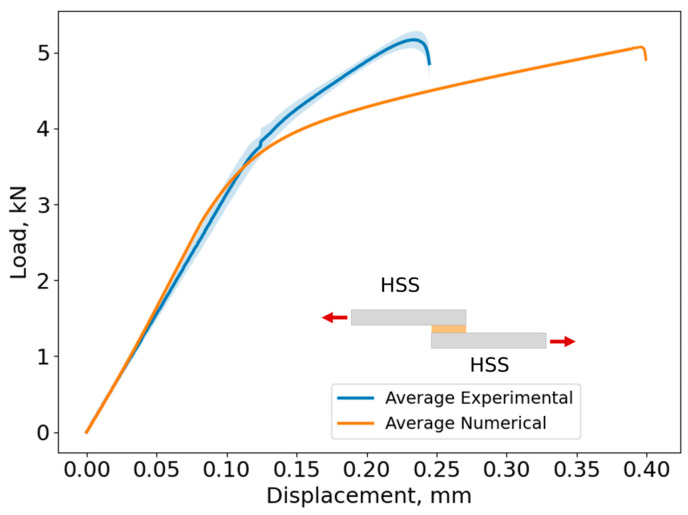
Load–displacement curves for numerical and experimental results of HSS–HSS SLJs of Adhesive A. The shaded area in the average experimental curve corresponds to standard deviation.

**Figure 27 biomimetics-10-00002-f027:**
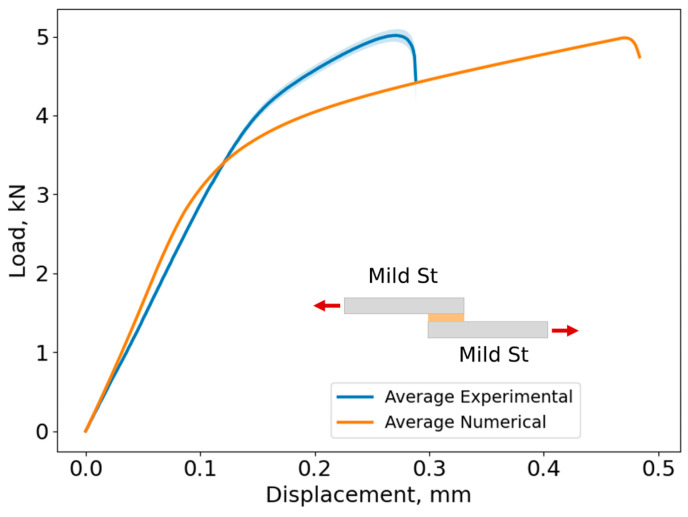
Load–displacement curves for numerical and experimental results of Mild St–Mild St SLJs of Adhesive A.

**Figure 28 biomimetics-10-00002-f028:**
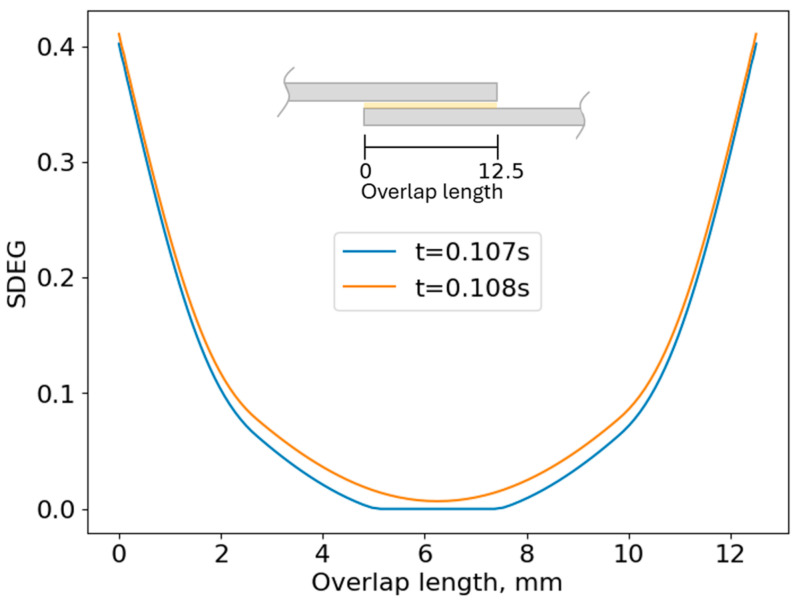
Damage evolution (SDEG) along the overlap length for the Mild St–Mild St SLJs for two consecutive time steps.

**Figure 29 biomimetics-10-00002-f029:**
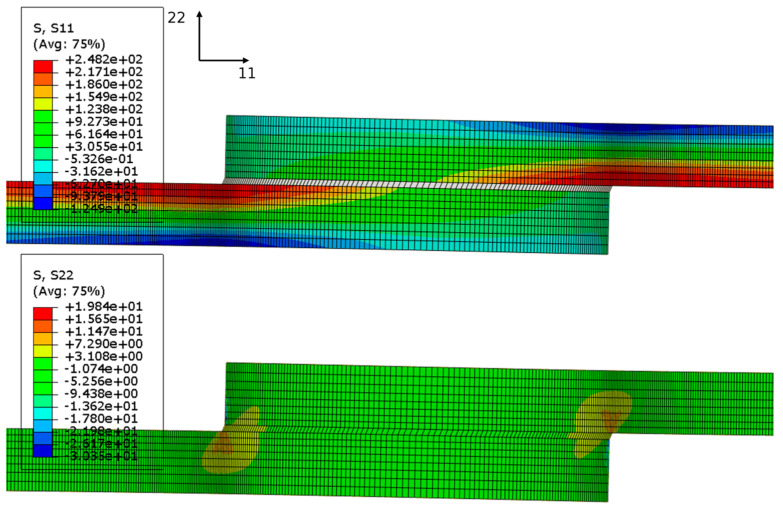
Stress field [MPa] in the loading direction (11) and in the normal direction (22).

**Table 1 biomimetics-10-00002-t001:** Tensile properties of Adhesives A and B.

Property	Units	Adhesive A	Adhesive B
Young’s modulus, *E*	MPa	572 ± 38	639 ± 68
Poisson’s ratio, ν	-	0.30 ± 0.01	0.34 ± 0.03
$\mathrm{Tensile}\;\mathrm{strength},\;\sigma_{f}$	MPa	12.5 ± 0.7	11.5 ± 0.7
$\mathrm{Tensile}\;\mathrm{strain}\;\mathrm{to}\;\mathrm{failure},\;\varepsilon_{f}$	%	85.0 ± 7.9	64.6 ± 6.0

**Table 2 biomimetics-10-00002-t002:** Shear properties of Adhesives A and B.

Property	Units	Adhesive A	Adhesive B
Shear modulus, *G*	MPa	211 ± 17	235 ± 63
$\mathrm{Shear}\;\mathrm{strength},\;\tau_{f}$	MPa	9.0 ± 0.2	8.8 ± 0.4
$\mathrm{Shear}\;\mathrm{strain}\;\mathrm{to}\;\mathrm{failure},\;\gamma_{f}$	%	61.8 ± 7.6	68.4 ± 6.9

**Table 3 biomimetics-10-00002-t003:** Summary of the mechanical properties of Adhesive A and Adhesive B.

Property	Units	Adhesive A	Adhesive B
Young’s modulus, *E*	MPa	572 ± 38	639 ± 68
Poisson’s ratio, ν	-	0.30 ± 0.01	0.34 ± 0.03
$\mathrm{Tensile}\;\mathrm{strength},\;\sigma_{f}$	MPa	12.5 ± 0.7	11.5 ± 0.7
$\mathrm{Tensile}\;\mathrm{strain}\;\mathrm{to}\;\mathrm{failure},\;\varepsilon_{f}$	%	85.0 ± 7.9	64.6 ± 6.0
Shear modulus, *G*	MPa	211 ± 17	235 ± 63
$\mathrm{Shear}\;\mathrm{strength},\;\tau_{f}$	MPa	9.0 ± 0.2	8.8 ± 0.4
$\mathrm{Shear}\;\mathrm{strain}\;\mathrm{to}\;\mathrm{failure},\;\gamma_{f}$	%	61.8 ± 7.6	68.4 ± 6.9
$\mathrm{Toughness}\;\mathrm{Mode}\;I,\;G_{IC}$	N/mm	1.3 ± 0.07	1.24 ± 0.15
$\mathrm{Toughness}\;\mathrm{Mode}\;\mathrm{II},\;G_{IIC}$	N/mm	6.0	6.0

## Data Availability

Data is contained within the article.

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
