# Peer review of "Exploring Adhesive Performance in Horseshoe Bonding Through Advanced Mechanical and Numerical Analysis"

_biomimetics, 2024, doi:10.3390/biomimetics10010002_

Round 1

Reviewer 1 Report

Comments and Suggestions for Authors

This paper aims to study the adhesive performance in horseshoe bonding through advanced mechanical and numerical analysis. The authors used different fracture specimens using two different adhesives. The experimental work is well reported and experimental data adequately analyzed. However, the current version is not suitable for being published in Biomimetics.

Tracking references for sections, figures, and citations is challenging, possibly due to errors in LaTeX processing.

Details about the test setup, particularly for the SLJ, are not provided.

For the dissimilar SLJ, selecting the same thickness for both adherends might induce out-of-plane bending. What was the reason for choosing equal thickness adherends?

Were tabs used during testing, and what material was used for the tabs?

Given the SLJ configuration shown in Figure 6, is there any effect of out-of-plane bending caused by loading eccentricity?

Additional details about the FEM are required, including mesh size, the CZM technique implemented in Abaqus, and surface linking methods, among others.

A comparison between the FEM results and the experimental findings is necessary.

It is recommended to include a stress field contour to highlight any potential loading eccentricity.

Additionally, analyze the mode mixity along the fracture process zone to determine whether it is purely Mode II or if there is some degree of mixity.

Reviewer 2 Report

Comments and Suggestions for Authors

This manuscript described an interesting problem, which is an alternative adhesive bonding solution compared to the traditional nail bonding. The background of the problem is well described. However, the novelty of the paper in terms of methodology and new findings are weak. Thus need to be revised.

1) As no direct comparison between traditional nail-based bonding and adhesive-based bonding is shown in the paper, it is hard to get any conclusion from this paper. As mentioned in introduction, this study aims to serve as a ground work. However, it is not clear from current content that what aspects of the adhesive need improve, in order to "instill the necessary confidence". Is it possible to make some comparison rather than just presenting the characterization of the adhesive bonding?

2) What failure mode is the adhesive bonding if using in horseshoe application? Authors listed the tests done by tensile specimens; TAST specimens; DCB specimens; SLJ specimens. Which test can most represent the real failure mode in horseshoe bonding? And what is the purpose of other tests if they are not directly relate to the problem?

3) The test in section 4.1.5 may not be properly setup, as it is hard to avoid load misalignment. This bending effect will cause the brittler HW plate much easier to break. A better setup is needed to completely eliminate the misalignment effect. 

4) In conclusion, authors mentioned that the next step is to incorporating the hoof and horseshoe geometry into the numerical model. However, what is the difficulty here? Why don't directly model the real geometry in this current study based on all the test data? That will give a much more compelling result.

5) Formatting issue: all of the references are missing in the current manuscript, please correct them.

Reviewer 3 Report

Comments and Suggestions for Authors

Section 3 ("Numerical Modeling") lacks sufficient explanation regarding the rationale for the selected cohesive law form (triangular for Mode I and trapezoidal for Mode II). It would be beneficial to provide a detailed justification as to why these forms are more effective in describing the adhesive behaviour than other approaches documented in the existing literature.

The text contains numerous references to "Error! Reference source not found.", which gives the impression that the results are unreliable. This requires correction to ensure the accurate presentation of the material.

Some tests employ only three samples, which is insufficient for ensuring the statistical reliability of the results. It would be prudent to increase the sample size.

The numerical calculations indicate discrepancies in the stiffness predictions for the HSS-HSS and mild steel-mild steel combinations. However, the text does not elucidate the reasons for these discrepancies nor potential methods to eliminate them.

It is necessary to provide explanations for Figures 17, 19, 22 and 23 on them with the text and arrows.

Round 2

Reviewer 2 Report

Comments and Suggestions for Authors

The revised version addressed my concern.